

# Dense GM-CSFRα-expressing immune infiltration is allied with longer survival of intrahepatic cholangiocarcinoma patients

Paksiree Saranaruk[1,2], Sakda Waraasawapati[3], Yaovalux Chamgramol[3], Kanlayanee Sawanyawisuth[1,2], Natnicha Paungpan[1,2,4], Narumon Somphud[1], Chaisiri Wongkham[1], Seiji Okada[4], Sopit Wongkham[1,2,4] and Kulthida Vaeteewoottacharn[1,2,4]

[1] Department of Biochemistry, Faculty of Medicine, Khon Kaen University, Khon Kaen, Thailand
[2] Cholangiocarcinoma Research Institute, Khon Kaen University, Khon Kaen, Thailand
[3] Department of Pathology, Faculty of Medicine, Khon Kaen University, Khon Kaen, Thailand
[4] Division of Hematopoiesis, Joint Research Center for Human Retrovirus Infection and Graduate School of Medical Sciences, Kumamoto University, Kumamoto, Japan

Corresponding author
Kulthida Vaeteewoottacharn,
kulthidava@kku.ac.th

## ABSTRACT

**Background.** Intrahepatic cholangiocarcinoma (iCCA) is a cancer arising from intrahepatic bile duct epithelium. An iCCA incidence is increasing worldwide; however, the outcome of the disease is dismal. The linkage between chronic inflammation and iCCA progression is well established, but the roles of granulocyte-macrophage colony-stimulating factor (GM-CSF) remain unrevealed. Thus, a better understanding of GM-CSF functions in CCA may provide an alternative approach to CCA treatment.

**Methods.** Differential *GM-CSF* and *GM-CSFRα* mRNA expressions in CCA tissues were investigated by Gene Expression Profiling Interactive Analysis (GEPIA) based on The Cancer Genome Atlas (TCGA) database. The protein expressions and localizations of GM-CSF and its cognate receptor (GM-CSFRα) in iCCA patients' tissues were demonstrated by the immunohistochemistry (IHC) techniques. The survival analyses were performed using Kaplan-Meier survival analysis with log-rank test and Cox proportional hazard regression model for multivariate analysis. The GM-CSF productions and GM-CSFRα expressions on CCA cells were assessed by ELISA and flow cytometry. The effects of GM-CSF on CCA cell proliferation and migration were evaluated after recombinant human GM-CSF treatment. The relationship between *GM-CSF* or *GM-CSFRα* level and related immune cell infiltration was analyzed using the Tumor Immune Estimation Resource (TIMER).

**Results.** GEPIA analysis indicated *GM-CSF* and *GM-CSFRα* expressions were higher in CCA tissues than in normal counterparts, and high *GM-CSFRα* was related to the longer disease-free survival of the patients ($p < 0.001$). IHC analysis revealed that CCA cells differentially expressed GM-CSF, while GM-CSFRα was expressed on cancer-infiltrating immune cells. The patient whose CCA tissue contained high GM-CSF expressed CCA, and moderate to dense GM-CSFRα-expressing immune cell infiltration (ICI) acquired longer overall survival (OS) ($p = 0.047$), whereas light GM-CSFRα-expressing ICI contributed to an increased hazard ratio (HR) to 1.882 (95% CI [1.077–3.287]; $p = 0.026$). In non-papillary subtype, an aggressive CCA subtype, patients with

light GM-CSFR$\alpha$-expressing ICI had shorter median OS (181 *vs.* 351 days; $p = 0.002$) and the HR was elevated to 2.788 (95% CI [1.299–5.985]; $p = 0.009$). Additionally, TIMER analysis demonstrated *GM-CSFR$\alpha$* expression was positively correlated with neutrophil, dendritic cell, and CD8+ T cell infiltrations, though it was conversely related to M2-macrophage and myeloid-derived suppressor cell infiltration. However, the direct effects of GM-CSF on CCA cell proliferation and migration were not observed in the current study.

**Conclusions**. Light GM-CSFR$\alpha$-expressing ICI was an independent poor prognostic factor for iCCA patients. Anti-cancer functions of GM-CSFR$\alpha$-expressing ICI were suggested. Altogether, the benefits of acquired GM-CSFR$\alpha$-expressing ICI and GM-CSF for CCA treatment are proposed herein and require elucidation.

## INTRODUCTION

Cholangiocarcinoma (CCA) is an epithelial cancer originating from the neoplastic transformation of bile duct lining cells, which the highest incidence has been reported in the Mekong subregion of Asia, including Thailand (*Banales et al., 2020*). According to anatomical classification, CCA is divided into intrahepatic (iCCA) and extrahepatic CCA (eCCA) (*Blechacz, 2017*). Based on histological characteristics, CCA is further classified into papillary and non-papillary subtypes (*Zen et al., 2006*). While the incidence of eCCA remains constant, the incidence of iCCA increases worldwide (*Blechacz, 2017*; *Khan, Tavolari & Brandi, 2019*). Despite a better prognosis of the papillary subtype, iCCA still has a dismal overall prognosis due to late diagnosis and high recurrence rate (*Banales et al., 2020*; *Blechacz, 2017*; *Zen et al., 2006*). The unique cause of liver fluke, *Opisthorchis viverrini*, infection is one contributing factor to the high number of cases (*Khan, Tavolari & Brandi, 2019*). Although targeted therapies against the fibroblast growth factor receptor (FGFR) and programmed death-ligand 1 (PD-L1) have shown promising results in clinical trials (*Banales et al., 2020*; *Rizzo, Ricci & Brandi, 2021a*; *Rizzo, Ricci & Brandi, 2021b*), the lack of these targets in fluke-associated CCA may hinder the clinical benefit in a specific population (*Kongpetch et al., 2020*; *Sangkhamanon et al., 2017*).

Chronic inflammation is closely related to CCA development and progression. Functions of certain cytokines are explored; CCA and cancer stromal cells secrete inflammatory cytokines, such as IL-1, IL-6, IL-8, IL-10, TNF-$\alpha$, and TGF-$\beta$, to promote cancer growth, metastasis, and immune evasion. Several studies target these cytokine-related signaling pathways as promising therapy (*Raggi, Invernizzi & Andersen, 2015*; *Roy, Glaser & Chakraborty, 2019*; *Vaeteewoottacharn et al., 2019*). GM-CSF is a hemopoietic growth factor that stimulates myeloid cell proliferation, differentiation, and survival. The GM-CSF cognate receptor (GM-CSFR), comprised of a ligand-binding subunit ($\alpha$) and a signaling subunit ($\beta$), is expressed on myeloid cells such as monocytes, macrophages, and dendritic cells (DC) (*Hamilton, 2020*). In a physiological state, GM-CSF is produced locally

in lung tissue to regulate alveolar macrophage phagocytosis (*Becher, Tugues & Greter, 2016*). It plays a crucial role in inflammatory-related diseases. GM-CSF and GM-CSFR are expressed in various cell types in response to inflammation (*Hamilton, 2020*). Regarding the cancer milieu, the controversial roles of GM-CSF were demonstrated. Cancer-promoting functions are established in breast, glioma, liver, and pancreatic cancers (*Bayne et al., 2012*; *Kohanbash et al., 2013*; *Lin et al., 2017*; *Reggiani et al., 2017*; *Revoltella, Menicagli & Campani, 2012*), whereas cancer-preventing roles are demonstrated in bladder, cervical, colon, esophageal, and prostate cancers (*Hori et al., 2019*; *Jiang et al., 2015*; *Nebiker et al., 2014*; *Wei et al., 2016*; *Zhang et al., 2017*). In CCA, there is a report demonstrating that cirrhosis-related iCCA-derived GM-CSF promotes intense neutrophil infiltration (*Sasaki et al., 2003*); however, the contradictory result is demonstrated in the spontaneous iCCA mouse model. Blockade of GM-CSF attenuated tumor-associated macrophages (TAMs) and facilitated cytotoxic T-cell infiltration (*Ruffolo et al., 2022*). Altogether, the importance of GM-CSF in iCCA remains unclear. Thus, GM-CSF and GM-CSFRα were the focus of this study.

The GM-CSF and GM-CSFRα expressions in patient iCCA tissues were investigated and correlated with the clinical parameters. The direct effects of GM-CSF on CCA cell proliferation and migration were assessed *in vitro*. The immune cell infiltration (ICI) profile related to *GM-CSF* and *GM-CSFRα* expressions in CCA was analyzed by a web-based tool, the Tumor Immune Estimation Resource (TIMER). Altogether, this study could better understand GM-CSF roles and their implications in iCCA.

## MATERIALS & METHODS

### Human iCCA tissues
Ninety-six formalin-fixed paraffin-embedded tissues were obtained with informed consent from the specimen bank of Cholangiocarcinoma Research Institute, Khon Kaen University, Khon Kaen, Thailand. Tissues were collected from 1998–2012 and were selected under the following criteria: (1) cancers were iCCA, (2) samples were from hepatic resection, (3) clinical data were available for analysis, and (4) patients had no known history of other cancers. Perioperative deaths or patients with an overall survival less than 30 days were excluded. The study protocol was approved by the Ethics Committee for Human Research of Khon Kaen University based on the Declaration of Helsinki (HE571283 and HE611034).

### Cell line and cell culture
Four CCA cell lines, KKU-055, KKU-100, KKU-213A, and KKU-213B were obtained from the Japanese Collection of Research Bioresources Cell Bank (Osaka, Japan) (*Sripa et al., 2005*; *Sripa et al., 2020*). Two metastatic cell lines, KKU-213L5 and KKU-214L5, were established as previously described (*Saentaweesuk et al., 2018*; *Uthaisar et al., 2016*). Cells were maintained in DMEM containing 10% FBS with 1% antibiotic-antimycotic in humidified 5% $CO_2$ at 37 °C. All cell culture-related reagents were obtained from Gibco (NY, USA).

## Antibodies and reagents

The sources of antibodies were as follows: rabbit anti-GM-CSF antibody was from Novus (NB600-632; Novus, St. Louis, MO, USA), mouse anti-GM-CSFRα was from Santa Cruz (4H1; TX, USA), anti-mouse and anti-rabbit EnVision-horseradish peroxidase (HRP)-conjugated antibodies were from DAKO (K4001 and K4003; DAKO, Glostrup, Denmark), phycoerythrin (PE)-conjugated anti-GM-CSFRα was from BioLegend (4H1; San Diego, CA, USA), PE-conjugated mouse IgG was from eBioscience (P3.6.2.8.1; eBioscience, San Diego, CA, USA). ELISA MAX$^{TM}$ Deluxe Set Human GM-CSF and recombinant human GM-CSF (rhGM-CSF) were from BioLegend. All chemicals were purchased from Sigma-Aldrich (St. Louis, MO, USA).

## Analysis of *GM-CSF* and *GM-CSFRα* mRNA expressions in CCA tissues

The *GM-CSF* and *GM-CSFRα* mRNA expressions in CCA tissues compared with their normal counterparts were investigated by Gene Expression Profiling Interactive Analysis (GEPIA, http://gepia.cancer-pku.cn/index.html), an online server for cancer and normal gene expression profiling analysis based on the Cancer Genome Atlas (TCGA) database (*Tang et al., 2017*). The RNA-Seq data were obtained from 36 CCA tissues and 9 adjacent normal tissues. *GM-CSF* and *GM-CSFRα* in CCA tissues were categorized into low and high expression by dichotomizing at the median.

## Immunohistochemistry

Immunohistochemical staining of GM-CSF and GM-CSFRα was performed as previously described (*Vaeteewoottacharn et al., 2019*) using rabbit anti-GM-CSF and mouse anti-GM-CSFRα antibodies. Immunoreactivity was developed using 3, 3′ diaminobenzidine. The signals were amplified using the corresponding EnVision-HRP system. The immunohistochemical evaluations were performed by two independent evaluators. GM-CSF expression levels of CCA cells were assessed by H-score (*Fitzgibbons et al., 2014*), while GM-CSFRα expression was categorized by the density of GM-CSFRα-expressing ICI in the cancer area into light and moderate to dense (*Wu et al., 2020*).

## GM-CSF ELISA

To determine GM-CSF concentration in CCA cultured media (CM), CM was collected at 24 h and determined GM-CSF concentration by GM-CSF ELISA kit following the manufacturer's instruction. The absorbance was measured at 450 nm by an iMark microplate reader (Bio-Rad, Hercules, CA, USA).

## Flow cytometry

GM-CSFRα expressions on surface CCA cell lines were evaluated by flow cytometry (LSR II flow cytometry; BD Biosciences, San Jose, CA, USA). Cells were stained by PE-conjugated anti-GM-CSFRα. PE-conjugated mouse IgG1 was used as isotype control. Data were analyzed by FlowJo software V.10.7.2 (Tree Star, Woodburn, OR, USA). GM-CSFRα expression levels were calculated as mean fluorescence intensity (MFI) of anti-GM-CSFRα-stained cells/MFI of isotype control-stained cells and were expressed as the relative MFI (*Vaeteewoottacharn et al., 2019*).

## MTT assay

The effects of GM-CSF on KKU-055 and KKU-213B cell numbers were determined by MTT assay. Cells were seeded at $2 \times 10^3$ cells per well onto a 96-well plate and treated with 0, 1, and 10 ng/ml rhGM-CSF for 24, 48, and 72 h. MTT solution was added to yield a final concentration of 0.5 mg/ml, and cells were incubated for an additional 3 h. Then formazan crystals were dissolved in acidified isopropanol (0.04 N HCl in isopropanol). The absorbance at 540 nm was determined by a Sunrise microplate reader (Tecan US, Inc., Morrisville, NC, USA).

## Scratch-wound assay

The scratch-wound assay was performed to determine the effect of GM-CSF on CCA migration. KKU-055 and KKU-213B were seeded into 24-well plates as a monolayer until confluent, and then the scratch was made using a sterile 200-μl tip, and cells were allowed to migrate in 0, 1, and 10 ng/ml rhGM-CSF containing media for 24 h. The images of wound closure area at 0, 6, 12, 18, and 24 h were taken by Nikon's DS-Fi2 camera connected to a Nikon Eclipse inverted microscope (Nikon Instruments Inc., Tokyo, Japan) and then calculated by ImageJ software V.1.48 as previous described (*Grada et al., 2017*). The percentage of wound closure was calculated as follows; (area at time 0 - area at an indicated time) $\times$ 100/area at time 0.

## The estimation of immune cell infiltration

The ICI in CCA tissues was analyzed by a web-based tool, the Tumor Immune Estimation Resource (TIMER2.0, http://timer.cistrome.org/), based on computational algorithms of the TCGA database (*Li et al., 2020*). The association between *GM-CSF* and *GM-CSFRα* mRNA expression levels and levels of ICI, including neutrophils, dendritic cells (DCs), all macrophages, M2 macrophages, myeloid-derived suppressor cells (MDSCs), and CD8+ T cells were estimated and plotted as correlation scatter plots. The correlation coefficients between variables were presented as rho values.

## Statistical analysis

Statistical analyses were performed using SPSS (V.26.0) (IBM, NY, USA) as follows: the associations between categorical variables were determined by Pearson's chi-square tests ($\chi^2$), Kaplan–Meier survival curves were compared using the log-rank test, and multivariate analyses using Cox proportional hazard models were performed to identify independent prognostic factors for survival of patients, then the backward selection was tested to avoid confounding factors (*Grant, Hickey & Head, 2019*). The quantitative data were presented as mean $\pm$ SD of three independent experiments unless otherwise specified. The multiple comparisons of means were analyzed by ANOVA while comparing two groups was performed by student's $t$-test using GraphPad Prism (V.7.0) (GraphPad Software, Inc., San Diego, CA, USA). A $p$-value less than 0.05 was considered of statistical significance.

**Table 1  Correlations of GM-CSF and GM-CSFRα expressions with clinical parameters of iCCA patients.**

| Variables | n (%) | GM-CSF expression | | $p^{\#}$ | GM-CSFRα expression | | $p^{\#}$ |
|---|---|---|---|---|---|---|---|
| | | Low | High | | Light | Moderate to dense | |
| **Age** (years-old) | | | | | | | |
| <57 | 48 (50.0) | 22 | 26 | 0.683 | 28 | 20 | 0.200 |
| ≥ 57 | 48 (50.0) | 24 | 24 | | 34 | 14 | |
| **Gender** | | | | | | | |
| Male | 58 (60.4) | 29 | 29 | 0.614 | 37 | 21 | 0.841 |
| Female | 38 (39.6) | 17 | 21 | | 25 | 13 | |
| **Histological subtype** | | | | | | | |
| Papillary | 35 (36.5) | 11 | 24 | 0.014* | 22 | 13 | 0.789 |
| Non-papillary | 61 (63.5) | 35 | 26 | | 40 | 21 | |
| **TNM stage (7th AJCC[##])** | | | | | | | |
| I | 6 (6.3) | 1 | 5 | 0.258 | 6 | 0 | 0.258 |
| II | 10 (10.4) | 5 | 5 | | 7 | 3 | |
| III | 23 (24.0) | 14 | 9 | | 15 | 8 | |
| IV | 57 (59.4) | 26 | 31 | | 34 | 23 | |
| **LN metastasis** ($n = 85$[**]) | | | | | | | |
| Negative | 50 (58.8) | 22 | 28 | 0.499 | 31 | 19 | 0.936 |
| Positive | 35 (41.2) | 18 | 17 | | 22 | 13 | |
| **Distant metastasis** ($n = 78$[**]) | | | | | | | |
| Negative | 68 (87.2) | 34 | 34 | 0.237 | 43 | 25 | 0.298 |
| Positive | 10 (12.8) | 7 | 3 | | 8 | 2 | |
| **GM-CSFRα** | | | | | | | |
| Moderate to dense | 34 (35.4) | 13 | 21 | 0.160 | | | |
| Light | 62 (64.6) | 33 | 29 | | | | |

**Notes.**

LN, lymph node.

[#]$p$-values were analyzed using Pearson's chi-square test.

[##]Tumor staging is classified according to the 7[th] AJCC system (*Edge & Compton, 2010*).

*$p < 0.05$.

[**]incomplete information.

## RESULTS

### Demographic characteristics of iCCA patients

The clinical data of 96 iCCA cases are shown in Table 1. The patient ages ranged from 33–76, with a median age of 57. Fifty-eight were males and 38 were females (a male-to-female ratio is 3:2). Histologically, 35 were papillary subtypes and 61 were non-papillary. Cancers were classified according to the 7[th] edition of the American Joint Committee in Cancer (AJCC) staging system (*Edge & Compton, 2010*). Six-point three percent was stage I, 10.4% was stage II, 24.0% was stage III, and 59.4% was stage IV. Thirty-five of 85 cases (41.2%) had lymph node (LN) metastasis while distant metastasis was observed in 10 of 78 cases (12.8%). The survival times calculated from surgery to death ranged from 41 to 2,509 days, with a median survival time of 302.

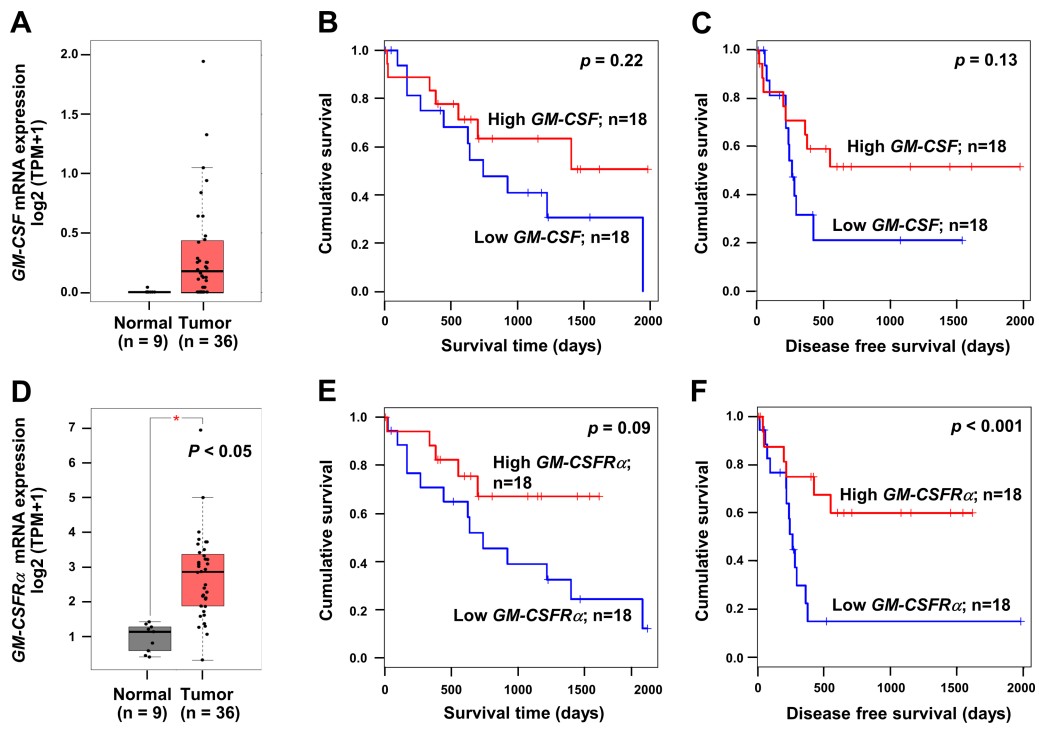

**Figure 1** **The mRNA expression levels and survival analyses of *GM-CSF* and *GM-CSFRα* in CCA tissues.** Box plots represent mRNA expression levels of (A) *GM-CSF* and (D) *GM-CSFRα* in patient CCA tissues (red) compared with normal counterparts (grey) based on TCGA database using a web-based tool, GEPIA analysis. Kaplan–Meier curves for overall survival and disease-free survival of low, and high (B-C) *GM-CSF* and (E–F) *GM-CSFRα*-expressing groups. *$p < 0.05$.

## Differential expressions of GM-CSF and GM-CSFRα were observed in iCCA tissues

To emphasize the significance of GM-CSF and GM-CSFRα on CCA progression, the mRNA expressions of both genes in CCA tissues were checked using a web-based tool, GEPIA, based on the TCGA database. *GM-CSF* and *GM-CSFRα* were highly expressed in cancer (Figs. 1A and 1D). Patients with high *GM-CSF* and *GM-CSFRα* in CCA tissues tended to have more prolonged overall survival (OS) and longer disease-free (DF) survival times than those with low expressions (Figs. 1B–1C and 1E–1F). Only higher *GM-CSFRα* expression in cancer tissues compared to the normal counterpart (Fig. 1D) and longer DF survival of patients with high *GM-CSFRα*- expressed CCA (Fig. 1F) showed statistical significance.

These data prompted determination of the GM-CSF and GM-CSFRα protein expressions in 96 iCCA tissues. GM-CSF expression was evaluated by H-score, while GM-CSFRα was graded by density. The representative figures are demonstrated in Figs. 2A and 2B. Immunohistochemistry staining revealed that CCA cells notably expressed GM-CSF (Fig. 2A), which was detectable in 95 cases (99%). On the other hand, GM-CSFRα was principally observed in non-CCA infiltrating cells localized in the peri-cancerous areas. These patterns morphologically resembled those previously described as immune cell infiltration (ICI)

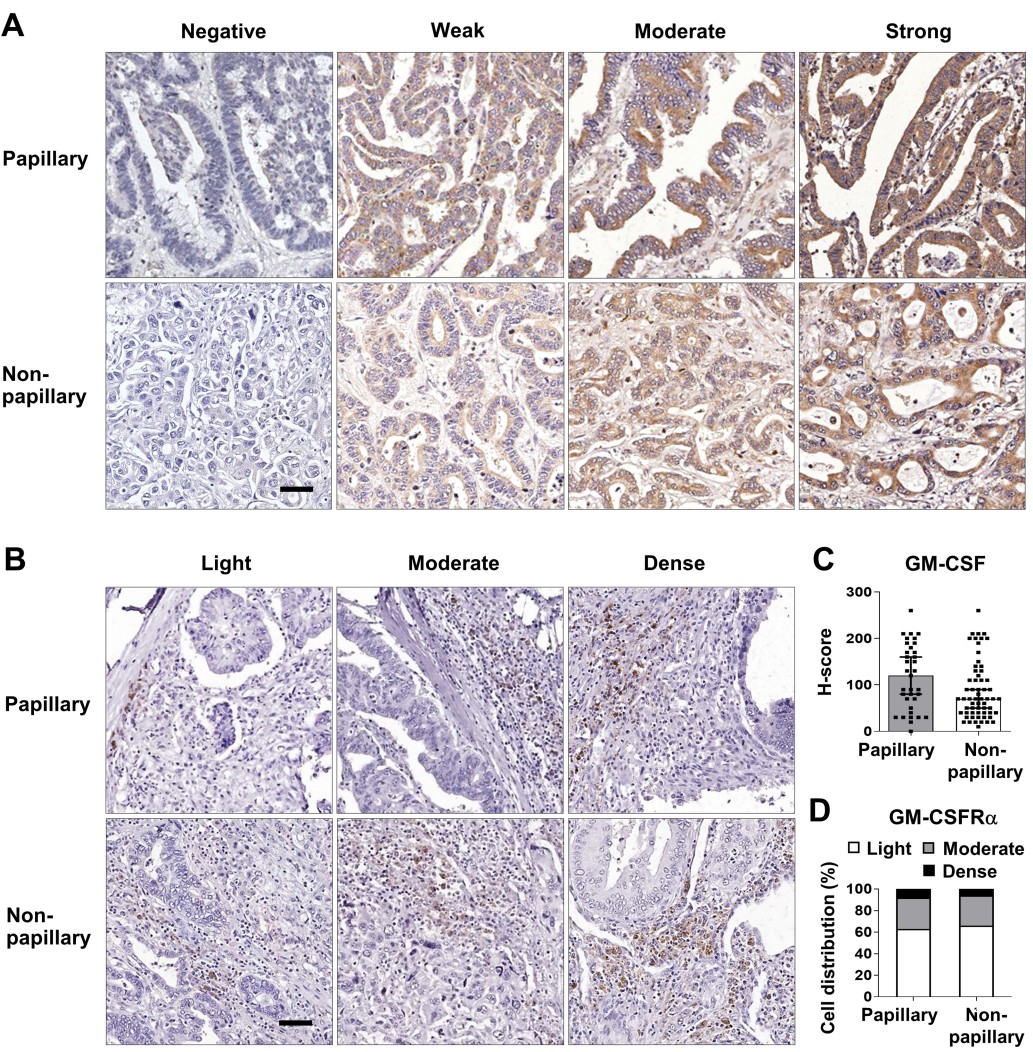

**Figure 2  GM-CSF and GM-CSFRα expressions in papillary and non-papillary subtypes of iCCA.** The representative immunohistochemistry staining of (A) GM-CSF and (B) GM-CSFRα-expressing ICI. GM-CSF in CCA cells are categorized into negative, weak, moderate, and strong staining and GM-CSFRα-expressing ICIs are classified as light, moderate, and dense infiltrations. Bar = 50 μm. The distributions of GM-CSF evaluated by H-score (C) and GM-CSFRα-expressing ICI densities (D) between two iCCA subtypes were compared.

(*Ino et al., 2013*). GM-CSFRα-expressed ICI was noticeable in all cases (Fig. 2B), while GM-CSFRα-expressing CCA cells were observed in 3 cases (3.1%) (Fig. S1).

Distinct histological subtypes of iCCA expressed GM-CSF differently (Fig. 2C). Therefore, further analyses were performed as the overall iCCA and subtype-specific. The median H-scores of GM-CSF in the overall iCCA, papillary and non-papillary subtypes were 80, 120, and 70. The median H-scores were used for dividing into low and high GM-CSF expressions. From a total of 96 cases, 50 (52.1%) were high GM-CSF (H-scores ≥ 80), while 46 (47.9%) were low (H-scores <80). Although the papillary subtypes expressed higher GM-CSF than the non-papillary ones, this difference was not significant (Fig. 2C).

In the papillary subtype, 17 cases of (48.6%, H-score <120) expressed low GM-CSF while 18 cases (51.4%, H-score ≥120) highly expressed GM-CSF. Among the non-papillary subtype, low and high GM-CSF expressions were observed in 27 (44.3%, H-score<70) and 34 cases (55.7%, H-score ≥70).

Light GM-CSFRα-expressing ICI was observed in 62 cases (64.6%), and moderate to dense infiltration was observed in 34 cases (35.4%; moderate = 27 cases, dense = 7 cases) (Table 1). The distributions of GM-CSFRα-expressing ICI were comparable between papillary and non-papillary subtypes (Fig. 2D). Light GM-CSFRα-expressing ICI was detected in 22 cases (62.9%) of papillary iCCA while light GM-CSFRα was noted in ICI of 40 non-papillary cases (65.6%).

## GM-CSFRα-expressing ICI was an independent prognostic factor for iCCA patients

Univariate analysis was performed to determine the correlation between GM-CSF or GM-CSFRα expressions and clinical parameters of the iCCA patients using $\chi^2$ test. The results demonstrated that the papillary subtype expressed higher GM-CSF (Table 1). Even though the patients with higher GM-CSF or moderate to dense GM-CSFRα seemed to have longer survival times, there were no statistically significant differences (Figs. 3A–3B); however, when the expressions of GM-CSF and GM-CSFRα were combined, the results showed high GM-CSF accompanied with moderate to dense GM-CSFRα-expressing ICI was correlated with longer median survival times (474 days) compared to one of these high expressions (329 days) or those who had decreased expressions of both proteins (209 days, $p = 0.047$) (Fig. 3C). Multivariate Cox regression analysis indicated that non-papillary subtype, TNM stage III, and light GM-CSFRα-expressing ICI were independent poor prognostic indicators (HR) = 2.130; 95% CI [1.046−4.337], $p = 0.037$; HR = 4.233, 95% CI [1.325–13.522], $p = 0.015$; and HR = 1.882, 95% CI [1.077–3.287], $p = 0.026$ (Table 2). It is worth noting that differential expressions of GM-CSF in distinct iCCA subtypes and GM-CSFRα in papillary iCCA were not correlated with the survival times of patients (Fig. S2).

It was reported that the different histological subtypes exhibited dissimilar disease progression; the papillary subtype seemed less aggressive and acquired a longer survival time (Zen et al., 2006). In agreement with a previous report, the longer median survival time was observed in patients with papillary iCCA that were recruited to the current study (456 vs. 236 days, $p < 0.001$) (Fig. 3D). Hence, different subtypes should be analyzed separately. The correlation between GM-CSF or GM-CSFRα expressions and clinical characteristics of the patients with distinct iCCA subtypes was performed. Longer median survival time of non-papillary iCCA was positively correlated with moderate to dense GM-CSFRα-expressing ICI (351 vs. 181 days, $p = 0.002$) (Fig. 3E). The multivariate Cox proportional hazards model indicated that light GM-CSFRα-expressing ICI increased risk of death to 2.788 times in non-papillary iCCA (95% CI [1.299–5.985], $p = 0.009$), while cancer stage III and lymph node metastasis increased risk of death 6.017 times, and 5.094 times in papillary iCCA (95%CI [1.394–25.959], $p = 0.016$ and 1.256–20.649, $p = 0.023$) (Table 3).

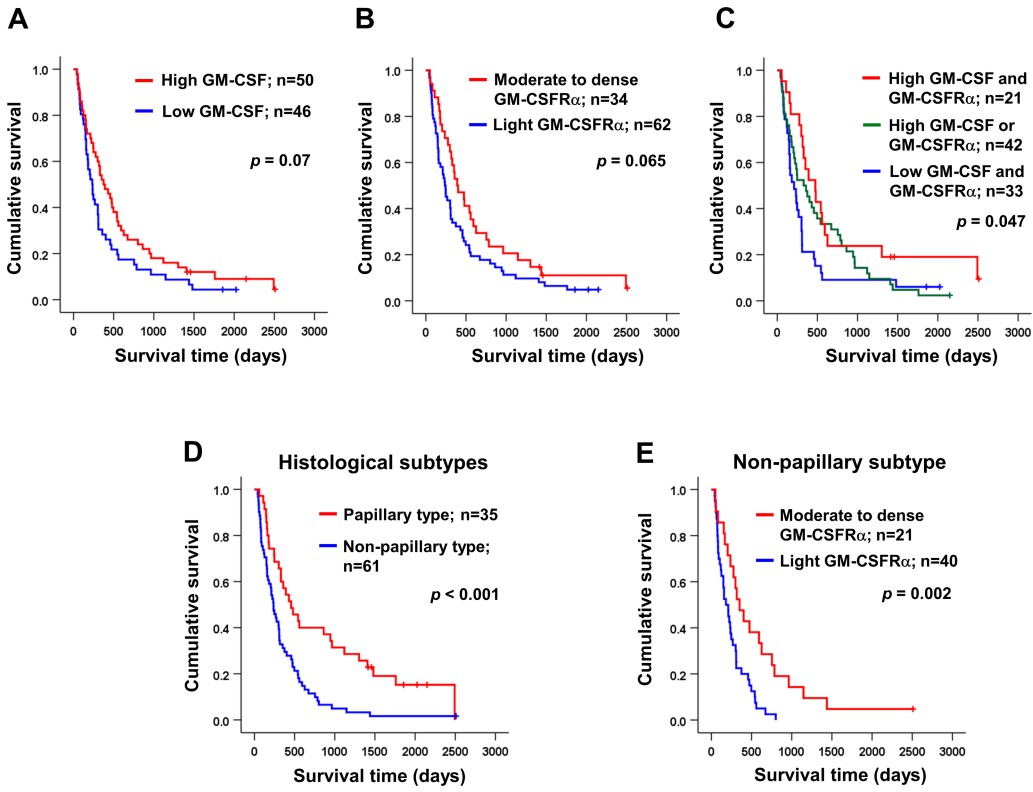

**Figure 3** **Kaplan–Meier survival analysis of iCCA patients with different GM-CSF and GM-CSFα expressions and distinct histological subtypes.** Kaplan–Meier survival analysis of iCCA patients with different (A) GM-CSF and (B) GM-CSFα expressions, and (C) combined GM-CSF and GM-CSFα expressions were demonstrated. (D) The median survival times of iCCA patients with papillary and non-papillary subtypes highlighted the favorable prognosis of papillary one. (E) High GM-CSFα-expressing ICI in non-papillary iCCA was related to longer survival times. Low GM-CSFα, light GM-CSFα-expressing ICI; high GM-CSFα, moderate to dense GM-CSFα-expressing ICI.

## GM-CSF had no direct effects on CCA cell proliferation and migration

To assess whether CCA cells produce GM-CSF, GM-CSF production was determined by detecting GM-CSF in conditioned media of CCA cells. The results showed GM-CSF was undetectable in KKU-055 and KKU-100 (<4 pg/ml), but in KKU-213A and KKU-213B, there were $84 \pm 6$ and $100 \pm 13$ pg/ml and were higher in metastatic cells, KKU-213L5 and KKU-214L5 ($349 \pm 21$ and $544 \pm 56$ pg/ml, Fig. 4A). To evaluate the possible roles of GM-CSF on CCA cells, the surface expression of its cognate receptor, GM-CSFα, on CCA cells was assessed by flow cytometry. GM-CSFα expressions on CCA cells varied from undetectable in KKU-213B to slight expression in KKU-213A and KKU-213L5 (1.11 and 1.12 times higher than MFI of isotype control). Moderate expressions were detected in KKU-055 and KKU-214L5 (2.44 and 2.17 times), while the highest expression was demonstrated in KKU-100 (9.93 times, Fig. 4B). Two CCA cell lines, KKU-055 and KKU-213B, were selected as representatives of GM-CSFα-expressing and GM-CSFα-non-expressing cells. The effects of GM-CSF on cell proliferation and migration were

**Table 2  Multivariate analysis using Cox proportional hazard regression model of iCCA clinical parameters.**

| Variables | | n | HR (95% CI) | p |
|---|---|---|---|---|
| Age (years-old) | <57 | 48 | 1 | |
| | ≥ 57 | 48 | 1.288 (0.722–2.299) | 0.392 |
| Gender | Male | 58 | 1 | |
| | Female | 38 | 0.683 (0.399–1.167) | 0.163 |
| Histological subtype | Papillary | 35 | 1 | |
| | Non-papillary | 61 | 2.130 (1.046–4.337) | 0.037[*] |
| TNM stage | I | 6 | 1 | |
| | II | 10 | 3.200 (0.770–13.296) | 0.109 |
| | III | 23 | 4.233 (1.325–13.522) | 0.015[*] |
| | IV | 57 | 2.185 (0.713–6.690) | 0.171 |
| LN metastasis | Negative | 50 | 1 | |
| | Positive | 35 | 1.671 (0.751–3.717) | 0.208 |
| Distant metastasis | Negative | 68 | 1 | |
| | Positive | 10 | 1.053 (0.458–2.423) | 0.903 |
| GM-CSF | High (≥ 80) | 50 | 1 | |
| | Low (<80) | 46 | 0.945 (0.525–1.701) | 0.850 |
| GM-CSFRα | Moderate to dense | 34 | 1 | |
| | Light | 62 | 1.882 (1.077–3.287) | 0.026[*] |

**Notes.**

HR, hazard ratio; CI, confidence interval; LN, lymph node.

[*]$p < 0.05$.

determined under rhGM-CSF treatment. Treatment with GM-CSF did not affect CCA cell proliferation and migration (Figs. 4C and 4D).

### *GM-CSF* and *GM-CSFRα* expression levels were correlated with specific ICI

As the moderate to dense ICI was correlated with longer survival time of specific iCCA, the potential immune cells were identified by an online tool, TIMER2.0. The infiltrating immune cells, which were reportedly associated with cancer-derived GM-CSF, including neutrophils, dendritic cells (DCs), all macrophages, M2 macrophages, and myeloid-derived suppressor cells (MDSCs), and CD8+ T cells, were selected (Figs. 5A and 5B). The results showed a positive correlation between specific groups of immune cells and *GM-CSF* or *GM-CSFRα* mRNA expressions. The *GM-CSF* expression level was positively correlated with infiltrations of DCs, all macrophages, and CD8+ T cells (Fig. 5A). Similarly, the higher *GM-CSFRα* expression was correlated with increased neutrophil, DC, and CD8+ T cell infiltrations. In contrast, the expression of *GM-CSFRα* was inversely correlated with M2 macrophage and MDSC infiltrations (Fig. 5B).
**Table 3    Multivariate Cox regression analysis of papillary and non-papillary iCCA subtypes.**

| Variables | n | Papillary subtype HR (95% CI) | p | n | Non-papillary subtype HR (95% CI) | p |
|---|---|---|---|---|---|---|
| **Age** | | | | | | |
| <57 years | 16 | 1 | | 32 | 1 | |
| ≥ 57 years | 19 | 1.145 (0.418–3.138) | 0.792 | 29 | 1.312 (0.550–3.131) | 0.540 |
| **Gender** | | | | | | |
| Male | 17 | 1 | | 41 | 1 | |
| Female | 18 | 2.169 (0.792–5.941) | 0.132 | 20 | 0.866 (0.397–1.888) | 0.717 |
| **Tumor stage** | | | | | | |
| I | 5 | 1 | | 1 | 1 | |
| II | 3 | 1.708 (0.147–19.855) | 0.669 | 7 | 4.801 (0.382–60.290) | 0.224 |
| III | 8 | 6.017 (1.394–25.959) | 0.016[*] | 15 | 4.277 (0.410–44.645) | 0.225 |
| IV | 19 | 1.127 (0.282–4.513) | 0.866 | 38 | 6.191 (0.597–64.247) | 0.127 |
| **LN metastasis** | | | | | | |
| Negative | 23 | 1 | | 27 | 1 | |
| Positive | 10 | 5.094 (1.256–20.649) | 0.023[*] | 25 | 0.649 (0.246–1.715) | 0.384 |
| **Distant metastasis** | | | | | | |
| Negative | | – | | 40 | 1 | |
| Positive | | | | 10 | 1.010 (0.412–2.478) | 0.982 |
| **GM-CSF** | | | | | | |
| High | 18 | 1 | | 34 | 1 | |
| Low | 17 | 1.747 (0.645–4.730) | 0.272 | 27 | 0.548 (0.275–1.093) | 0.088 |
| **GM-CSFRα** | | | | | | |
| Moderate to dense | 13 | 1 | | 21 | 1 | |
| Light | 22 | 1.629 (0.570–4.660) | 0.363 | 40 | 2.788 (1.299–5.985) | 0.009[*] |

**Notes.**
HR, hazard ratio; CI, confidence interval; LN, lymph node.
[*]$p < 0.05$.

# DISCUSSION

To date, surgical resection with a free surgical margin is the only potentially curative treatment for CCA, but the number of candidates is limited due to high metastasis (*Banales et al., 2020*; *Bridgewater et al., 2014*). Although chemotherapy has been proposed (*Valle et al., 2014*), the optimal approaches are still urgently needed to improve OS. Recently, immunotherapy and regulation of cytokine signaling have provided promising strategies for CCA treatment (*Guo et al., 2021*; *Nguyen et al., 2021*; *Panya et al., 2018*; *Vaeteewoottacharn et al., 2019*; *Yamanaka et al., 2020*) but the heterogeneous nature of the disease contributes to the requirement of target assessment in the specific population (*Kongpetch et al., 2020*; *Sangkhamanon et al., 2017*). GM-CSF is the most common immunostimulatory cytokine used in clinical vaccine trials (*Cuzzubbo et al., 2020*). The efficacy is still debated because GM-CSF activates both anti-cancer immunity and recruitment of cancer-promoting immune cells depending on the subsets of ICI in the cancer microenvironment (*Berraondo et al., 2019*; *Cuzzubbo et al., 2020*; *Garner & de Visser, 2020*; *Kumar et al., 2022*). Herein, the

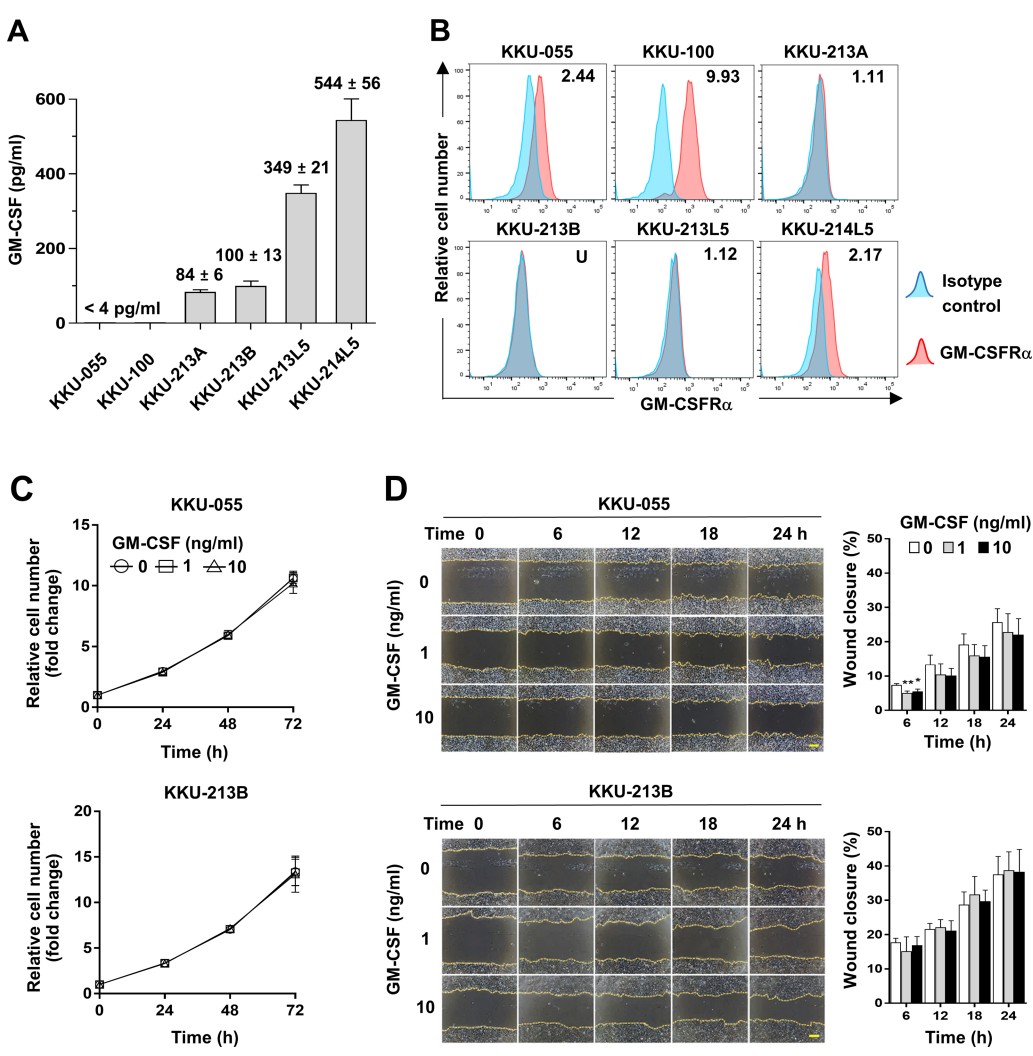

**Figure 4** **The effects of GM-CSF on CCA cell proliferation and migration.** (A) GM-CSF in conditioned media of six CCA cell lines was determined by ELISA and levels are presented as mean ±SD in pg/ml. (B) The surface GM-CSFRα expression (red) was measured by flow cytometry. MFI of isotype-stained cells (blue) is served as an internal control and used for normalization. The relative MFI of GM-CSFRα is shown on the upper right corner of each cell line. KKU-055 and KKU-213B were treated with 0, 1 and 10 ng/ml rhGM-CSF at indicated times to reveal the effects of GM-CSF on CCA cell proliferation (C) and migration (D). Data are presented as mean ±SD from 3 independent experiments, Bar = 200 μm.

potential functions of the inflammatory cytokine GM-CSF and its receptor, GM-CSFRα, in CCA were investigated.

The elevated *GM-CSF* and *GM-CSFRα* mRNA expressions in CCA tissues compared with normal counterparts by GEPIA analysis were first demonstrated. Patients with high *GM-CSFRα* appeared to have longer DFS. The levels of GM-CSF and GM-CSFRα in Thai iCCA patient tissues were determined by immunohistochemistry staining. GM-CSF was differentially expressed in CCA cells, whereas GM-CSFRα was observed primarily in ICI. The high expressions of GM-CSF in CCA and increased GM-CSFRα-expressed ICI seemed

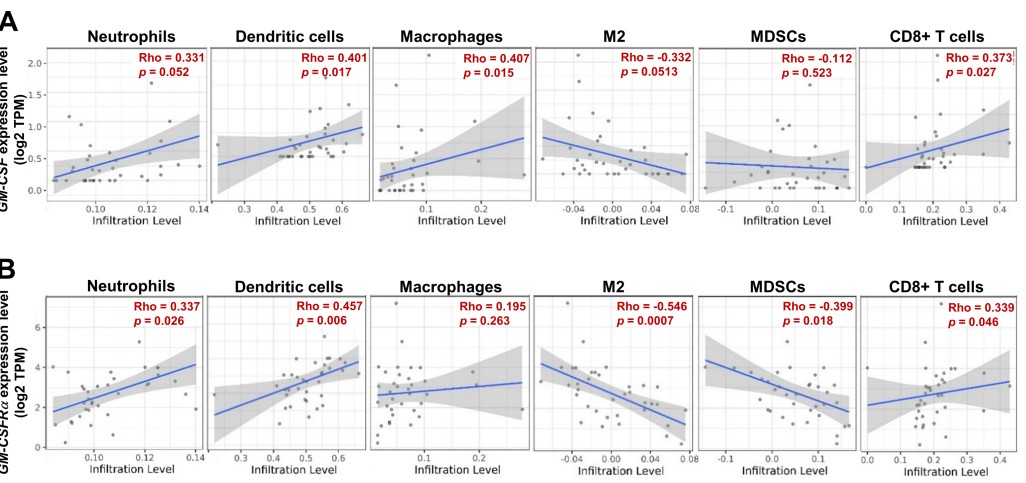

**Figure 5** **The correlation between GM-CSF and GM-CSFRα expressions and specific subsets of ICI.**
ICI in CCA tissues is demonstrated using a scatter plot in Tumor Immune Estimation Resource (TIMER)
2.0 using The Cancer Genome Atlas (TCGA) database. The correlations between *GM-CSF* (A) and
*GM-CSFRα* expression levels (B) and neutrophils, dendritic cells, all macrophages (macrophages), M2
macrophage (M2), monocyte-derived suppressing cells (MDSCs) and CD8+ T cell infiltrations. Rho
values represent the correlation coefficient.

to benefit CCA patients, particularly the worse prognostic subtype, the non-papillary one,
demonstrated by longer survival times (*Zen et al., 2006*). It is worth mentioning that the
characteristics of the iCCA patients in the current study represented a heterogeneous patient
population in a liver-fluke endemic area (*Kamsa-Ard et al., 2021*). Using multivariate
Cox regression analysis with backward selection, light GM-CSFRα-expressing ICI was
emphasized as an independent unfavorable prognostic factor for iCCA patients. Additional
investigation using TIMER2.0 suggested anti-CCA functions of GM-CSF and GM-CSFRα
as high *GM-CSF* and *GM-CSFRα* were associated with high neutrophil, DC, and CD8+
T cell infiltrations. Simultaneously, *GM-CSFRα* was negatively related to densities of
cancer-promoting immune cells (M2 macrophages and MDSCs).

In agreement with the present study, anti-cancer roles of GM-CSF are shown in
colon (COLO 205) (*Nebiker et al., 2014*), prostate (patient tissues) (*Wei et al., 2016*), CCA
(KKU-213A) (*Panya et al., 2018*), bladder (MGHU3, UMUC3) (*Hori et al., 2019*), cervical
(patient tissues) (*Jiang et al., 2015*) and esophageal cancers (Eca-109, EC9706) (*Zhang
et al., 2017*), through activations of CD16-positive monocytes and effector T-cells, but
suppressions of M2-mediated angiogenesis and pro-inflammatory mediator productions
(*e.g.*, cyclooxygenase-2 and inducible nitric oxide synthase). On the contrary, cancer-
derived GM-CSF promoting immunosuppressive cell infiltration is reported in the iCCA
mouse model (*Ruffolo et al., 2022*). The similar phenomenon was observed in other cancers
in which the low dose of GM-CSF provides benefit to the OS of the patient while the high
dose yields opposite results (*Hodi et al., 2014*; *Parmiani et al., 2007*). Thus, the adjustment
of GM-CSF delivery to cancer patients might be required in cancer tissues with differential
GM-CSF expressions.

The autocrine effect of GM-CSF on cancer growth promotion is also reported in GM-CSFR-expressing glioma cells (*Revoltella, Menicagli & Campani, 2012*), but the direct effects of exogenous GM-CSF on GM-CSFRα-expressing iCCA cell properties were not detected in the present study. These results implied GM-CSF functions on ICI in the CCA microenvironment. These results implied the indirect effects of GM-CSF on CCA progression. The effects on immune cells were speculated.

To determine the involvement of ICI in GM-CSF-expressing CCA, the analysis using TIMER2.0 suggested that high *GM-CSF*-expressed CCA was associated with DC macrophage, and CD8+ T cell infiltrations, and *GM-CSFRα* expression was allied to neutrophil, DC, and CD8+ T cell infiltrations. Increased *GM-CSFRα* was negatively correlated with the levels of M2-macrophage and MDSC in the CCA microenvironment. In CCA, the increased number of mature DCs correlates with T cell infiltration and a lower incidence of lymph node metastasis in patients (*Loeuillard et al., 2019*), whereas the immunosuppressive cells such as TAMs and MDSCs promote CCA growth *via* impairing cytotoxic T cell responses (*Loeuillard et al., 2020*; *Zhou et al., 2021*). Related to TIMER2.0 results, it is consistent with previous studies in which high GM-CSF is related to neutrophil infiltration in iCCA tissues (*Sasaki et al., 2003*), and GM-CSF promotes cytotoxic T cell activity against CCA cells (*Panya et al., 2018*). Additionally, these authors previously demonstrated that GM-CSF and M-CSF treatments to primary monocytes induce monocyte-derived macrophages (MDMs) and co-culture between MDMs and CCA cell line KKU-213A promotes cancer cell phagocytosis (*Vaeteewoottacharn et al., 2019*).

Altogether, GM-CSF in an iCCA setting may promote the anti-cancer immune response. This study highlights that GM-CSFRα-expressing ICI might be an independent good prognostic factor for iCCA patients. The functions of GM-CSF on GM-CSFRα-expressing cells, including monocyte, macrophage, and DC, are suggested herein. Further studies to identify the populations of GM-CSFRα-expressing immune cells and the direct effects of iCCA-derived GM-CSF on specific immune cell recruitments and functions are required. The advancement of single-cell analysis might provide the comprehensive detail regarding principle immune cells and their roles in GM-CSF high- and low-expressing CCA tissues (*Shi et al., 2022*); however, it is not possible in the KKU setting as all clinical samples in the present work are formalin-fixed paraffin-embedded CCA tissues.

## CONCLUSIONS

The expressions of GM-CSF and its receptor, GM-CSFRα, were investigated in iCCA. The ICI profile implied that GM-CSFRα expression is positively correlated with neutrophil, DC, and CD8+ T cell infiltrations but not the immunosuppressive cells. Moderate to dense GM-CSFRα-expressing ICI is the independent good prognostic factor for the patient's survival. Additional studies are required to support the advantages of GM-CSF and the contributions of GM-CSFRα-expressing immune cells in CCA treatment.

## ACKNOWLEDGEMENTS

The authors would like to acknowledge Prof. James A Will for editing the manuscript *via* Publication Clinic KKU, Thailand. We would like to thank Cholangiocarcinoma Research Institute (CARI), Khon Kaen University, Thailand for CCA tissues.

### Funding

This study was supported by Grant from the Program Management Unit for Human Resources & Institutional Development, Research and Innovation, Khon Kaen University, Thailand (grant #630000050061 to Kulthida Vaeteewoottacharn). Kulthida Vaeteewoottacharn was also supported by the Thailand Research Fund (Grant Number: RSA6180068). The funders had no role in study design, data collection and analysis, decision to publish, or preparation of the manuscript.

### Grant Disclosures

The following grant information was disclosed by the authors:
Grant from the Program Management Unit for Human Resources & Institutional Development, Research and Innovation, Khon Kaen University, Thailand: 630000050061.
Thailand Research Fund: RSA6180068.

### Competing Interests

The authors declare there are no competing interests.

### Author Contributions

- Paksiree Saranaruk conceived and designed the experiments, performed the experiments, analyzed the data, prepared figures and/or tables, authored or reviewed drafts of the article, and approved the final draft.
- Sakda Waraasawapati conceived and designed the experiments, performed the experiments, prepared figures and/or tables, authored or reviewed drafts of the article, and approved the final draft.
- Yaovalux Chamgramol conceived and designed the experiments, authored or reviewed drafts of the article, and approved the final draft.
- Kanlayanee Sawanyawisuth conceived and designed the experiments, authored or reviewed drafts of the article, and approved the final draft.
- Natnicha Paungpan analyzed the data, authored or reviewed drafts of the article, and approved the final draft.
- Narumon Somphud analyzed the data, authored or reviewed drafts of the article, and approved the final draft.
- Chaisiri Wongkham conceived and designed the experiments, authored or reviewed drafts of the article, and approved the final draft.
- Seiji Okada analyzed the data, authored or reviewed drafts of the article, and approved the final draft.

- Sopit Wongkham analyzed the data, authored or reviewed drafts of the article, and approved the final draft.
- Kulthida Vaeteewoottacharn conceived and designed the experiments, performed the experiments, analyzed the data, prepared figures and/or tables, authored or reviewed drafts of the article, and approved the final draft.

## Human Ethics

The following information was supplied relating to ethical approvals (i.e., approving body and any reference numbers):

The Ethics Committee for Human Research of Khon Kaen University based on the Declaration of Helsinki (HE571283 and HE611034)

## Data Availability

The raw data are available in the Supplemental Files.

## Supplemental Information

Supplemental information for this article can be found online at http://dx.doi.org/10.7717/peerj.14883#supplemental-information.

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
