# Peer review of "Dense GM-CSFRα-expressing immune infiltration is allied with longer survival of intrahepatic cholangiocarcinoma patients"

_PeerJ, doi:10.7717/peerj.14883_

## Round 0.1 · original submission · Major Revisions

Dear Dr. Vaeteewoottacharn,,

Thank you for submitting your manuscript "Dense GM-CSFRα -expressing immune infiltration is allied with longer survival of intrahepatic cholangiocarcinoma patients" to PeerJ. We have now received reports from the reviewers and, after careful consideration internally, we have decided to invite a major revision of the manuscript.

As you will see from the reports copied below, the reviewers raise important concerns regarding some experimental validation and the limitations of the study. We find that these concerns limit the strength of the study, and therefore we ask you to address them with additional work. Without substantial revisions, we will be unlikely to send the paper back for review.

If you feel that you are able to comprehensively address the reviewers’ concerns, please provide a point-by-point response to these comments along with your revision. Please show all changes in the manuscript text file with track changes or color highlighting. If you are unable to address specific reviewer requests or find any points invalid, please explain why in the point-by-point response.

Reviewer 1 has requested that you cite specific references. You may add them if you believe they are especially relevant. However, I do not expect you to include these citations, and if you do not include them, this will not influence my decision.

Thanks

Abhishek Tyagi, PhD
Academic Editor,
PeerJ

Reviewer 1 ·

Basic reporting

An interesting study. Major changes needed.

Experimental design

See below. Major changes needed.

Validity of the findings

Dear Editor, thank you so much for inviting me to revise this manuscript.

This study addresses a current topic.

The manuscript is quite well written and organized. English should be improved.
Figures and tables are comprehensive and clear.
The introduction explains in a clear and coherent manner the background of this study.

We suggest the following modifications:
- Introduction section: although the authors correctly included important papers in this setting, we believe the changing, evolving treatment scenario for CCA patients should be better discussed and some recently published papers added within the introduction ( PMID: 33645367; PMID: 32396398 ; PMID: 32994319 ; PMID: 33611090 ), only for a matter of consistency. We think it might be useful to introduce the topic of this interesting study.
- Methods and Statistical Analysis: nothing to add.
- An important bias is due to selection of patients. In fact, the authors included a widely heterogeneous patient population, in terms of disease stage and several other variables and factors. This is something that seriously limits the importance of the results of the current study.
- Discussion section: Very interesting and timely discussion. Of note, the authors should expand the Discussion section, including a more personal perspective to reflect on. For example, they could answer the following questions – in order to facilitate the understanding of this complex topic to readers: what potential does this study hold? What are the knowledge gaps and how do researchers tackle them? How do you see this area unfolding in the next 5 years? We think it would be extremely interesting for the readers.

However, we think the authors should be acknowledged for their work. In fact, they correctly addressed an important topic, the methods sound good and their discussion is well balanced.

One additional little flaw: the authors could better explain the limitations of their work, in the last part of the Discussion.

We believe this article is suitable for publication in the journal although major revisions are needed. The main strengths of this paper are that it addresses an interesting and very timely question and provides a clear answer, with some limitations.

We suggest a linguistic revision and the addition of some references for a matter of consistency. Moreover, the authors should better clarify some points.

Additional comments

nothing to add

Reviewer 2 ·

Basic reporting

1. Line29: ‘An iCCA incidence is increasing worldwide; however, the outcome of ’. Meaning of this sentence is not clear. Please revise the sentence.
2. Line48: ‘The patient’ should be ‘Patients’
3. Line230: 'However, only GM-CSFRa expression and positive correlation between GM-CSFRa and DF survival time'. Please revise this sentence to convey clear information.
4. Line254 ‘papillary subtype is expressed higher’, please correct the expression.
5. Line255 ‘GM-CSFa’ should be GM-CSFRa.
6. Line286: 'the cognate receptor, GM-CSFRa, expressions on CCA cells were assessed'. Please revise this sentence to become more concisely.
7. Line296: 'GM-CSF and GM-CSFRa' expression levels correlated with specific ICI.' should be: 'GM-CSF and GM-CSFRa' expression levels are correlated with specific ICI'
8. The overall writing should still be polished to avoid grammatical errors and increase readability.

Experimental design

1. The authors performed IHC on CCA patient tissues and declared those are GM-CSFRa positive ICI (Figure2B), which is arbitrary. According to Figure4B, some CCA cell lines expressed GM-CSFRa, so it is likely that CCA tumor cells also express GM-CSFRa. It may not appropriate to say GM-CSFRa positive cells are all ICI.

2. The experiment design for Figure4 is not sufficient to conclude the effect of GM-CSF-GM-CSFRa axis on CCA cells. Since GM-CSF could affect tumor cells through immune cells (such as macrophages), the authors can design a co-culture experiment to see whether adding GM-CSF inhibit the proliferation or promoting apoptosis of tumor cell lines in presence of immune cells. The authors could also knock-down GM-CSFRa in CCA cell lines to see if cell proliferation or migration will be altered.

3. The authors showed higher GM-CSFRa is associated with better overall survival rate in CCA patients and pro-inflammatory immune cell infiltration. However, since Figure4 did not show any mechanism, the author could compare the expression of immune check-point markers (such as PD1) between GM-CSFRa negative versus positive cells in order to provide potential mechanism.

Validity of the findings

no comment

Additional comments

no comment

---

## Round 0.2 · accepted · Accept

Dear Dr. Vaeteewoottacharn,

We are delighted to accept your manuscript, entitled "Dense GM-CSFRα -expressing immune infiltration is allied with longer survival of intrahepatic cholangiocarcinoma patients," for publication in PeerJ.

Thank you for choosing to publish your interesting work with us.


With kind regards,
Abhishek Tyagi
Academic Editor, PeerJ

Reviewer 1 ·

Basic reporting

The authors addressed all the queries and issues we raised.
We recommend Acceptance.

Experimental design

The authors addressed all the queries and issues we raised.
We recommend Acceptance.

Validity of the findings

The authors addressed all the queries and issues we raised.
We recommend Acceptance.

Additional comments

The authors addressed all the queries and issues we raised.
We recommend Acceptance.